# Multi-Sector Assessment and Client-Perception of Social Need at Long-Term Follow-Up of a Group-Randomized Trial of Community-Engaged Collaborative Care for Adults with Depression

**DOI:** 10.3390/ijerph191610212

**Published:** 2022-08-17

**Authors:** Nicolás E. Barceló, Enrico G. Castillo, Roya Ijadi-Maghsoodi, Nichole Goodsmith, Lingqi Tang, David Okikawa, Felica Jones, Pluscedia Williams, Christopher Benitez, Bowen Chung, Kenneth B. Wells

**Affiliations:** 1UCLA National Clinician Scholars Program, Los Angeles, CA 90024, USA; 2Center for Social Medicine and Humanities, UCLA Jane and Terry Semel Institute for Neuroscience and Human Behavior, Los Angeles, CA 90024, USA; 3Psychiatry and Biobehavioral Sciences, UCLA David Geffen School of Medicine, Los Angeles, CA 90095, USA; 4VA Health Service Research and Development Center for the Study of Healthcare Innovation, Implementation and Policy, VA Greater Los Angeles Healthcare System, Los Angeles, CA 90073, USA; 5Center for Health Services and Society, UCLA Jane and Terry Semel Institute for Neuroscience and Human Behavior, Los Angeles, CA 90024, USA; 6UCLA David Geffen School of Medicine, Los Angeles, CA 90095, USA; 7Healthy African American Families II, Los Angeles, CA 90008, USA; 8The Practice of Community Faculty, Charles R. Drew University of Science and Medicine, Los Angeles, CA 90059, USA; 9Lundquist Institute, Harbor-UCLA Medical Center, Los Angeles, CA 90502, USA; 10Emergency Medicine Palliative Care Access (EMPallA), NYU—Langone Health Care, New York City, NY 10016, USA; 11Los Angeles County Department of Mental Health, Los Angeles, CA 90020, USA; 12Department of Psychiatry, Harbor-UCLA Medical Center, Los Angeles Biomedical Research Institute, Los Angeles, CA 90502, USA; 13Health Policy and Management, UCLA Fielding School of Public Health, Los Angeles, CA 90095, USA; 14Greater Los Angeles Veterans Administration Health System, Los Angeles, CA 90073, USA

**Keywords:** social determinants of health, quality improvement, patient experience, depression, mental health equity

## Abstract

Understanding client perceptions of need for underlying social determinant support may improve services for depression care. This secondary analysis examines perceptions of “social needs” related to housing and employment, financial, and legal (EFL) concerns among individuals with depression. Data were analyzed from Community Partners in Care, a randomized comparative effectiveness trial of multi-sector collaborative care for depression among a sample of people who were predominantly racial/ethnic minorities and low-income. Adults with depression (*n* = 980) in both interventions were surveyed at 36-month follow-up for (1) being asked about and (2) having social needs for housing or EFL concerns. In multivariate models, life difficulty and mental health visits in non-healthcare sectors predicted being asked about housing and EFL. Lower social determinants of health-related life satisfaction increased the odds of having unmet housing and EFL needs. These findings underscore the role of non-healthcare organizations as community resources for depression care and in screening and addressing social needs.

## 1. Introduction

Via processes of marginalization and divestment, poor communities of color in the United States face a disproportionate burden of structurally and socially determined barriers to health. This disproportionate burden of social inequity fuels mental health inequities in prevalence, severity, service utilization, and outcomes related to depression [1,2,3]. In turn, interventions to address the structural forces affecting mental health are poised to alleviate suffering for those with mental illness, protect the population’s mental health, and reduce inequities in behavioral health across population groups [4,5,6].

Recent literature exploring health-related societal factors offers a useful framework for analysis [7]. Social determinants of health, the macro and population-level processes influenced by policy and social norms that shape numerous life domains, confer either advantages to health or serve as barriers to life and longevity [4,5,6,7,8]. Social risk factors is a concept extending from social determinants of health referring to tangible, adverse social exposures experienced at the individual level [9,10]. These social risk factors, such as homelessness or unemployment, are commonly screened at an individual level at the time of service use and may be indicative of cumulative impact of the social determinants of health throughout the life course [7,8,9]. Social needs are the social risk factors that individual patients/clients identify as a concern and/or prioritize for social services intervention [11]. In this way [12], assessment of social needs is consistent with patient-centered approaches to delivery [13] of services related to social concerns impacting health.

Among interventions targeting the impact of social factors on health, substantial attention has been paid to the social determinants of health and social risk factors [10]. Less is known, however, regarding how social risk factor screening influences client perceptions of social need and/or how the screening context (in healthcare vs. non-healthcare agencies) influences client report of these social needs. For these reasons, and to improve the impact of social risk factor interventions, clients’ preferences and experiences should be investigated to uncover social needs that are most relevant and in need of intervention. Further, despite recognition that individuals from under-resourced communities who face social risk factors are likely to depend on multiple service agencies operating in disjointed systems of care [14], a majority of available studies related to interventions addressing social risk typically focus on the institutional efforts of single healthcare systems and organizations [15]. These trends—a limited emphasis on individual service-user prioritization of social need (relative to social risk factor identification) and a general focus on organizational performance for individual institutions—suggest a need to direct attention to more client-centered experiences.

Characterizing the client experience of social risk factor screening across sectors may be essential to efforts to realize the benefit of multisector networks and to promote enhanced coordination of clients with intersecting behavioral health, medical, and social needs [14]. Similarly, understanding the interplay of individual and institutional factors that influence clients’ perception and reporting of social needs may better enable multisector networks to identify barriers impeding engagement of services while maintaining client and community perspectives as the center of their work. The objective of this exploratory analysis is to identify factors associated with being asked about social need and having the perception of unmet social need among a population of adults with depression from under-resourced communities.

To accomplish this objective, we conducted an exploratory secondary analysis of long-term follow-up data (from 36 months survey outcomes) from Community Partners in Care (CPIC) [16], a large, community-partnered comparative effectiveness trial of multi-sector collaborative care for depression. From this sample of adults with depression facing high levels of social marginalization, we examined clients’ report of being asked about social need across multiple sectors and their perceptions of unmet social need and modeled patient- and systems-level factors associated with each outcome. Utilizing a socio-ecological framework [17], we hypothesized that both patient- (including but not limited to individual clinical symptoms, service-use patterns, experiences of barriers to care, report of lifetime stressful events, and measures of recent life satisfaction) and system-level factors would predict report of being asked about social needs and perceptions of unmet social need at follow-up (3 years). Additionally, compared to other time periods, we hypothesized that experiences in the 6 months prior to the assessment at 3 years would be particularly predictive of reported social needs.

## 2. Materials and Methods

### 2.1. Community Partners in Care

The Community Partners in Care (CPIC) evaluation of expanded collaborative care services for depression took place in two under-resourced, minority-majority communities in Los Angeles [16]. In addition to mental health and substance use needs, participants enrolled in CPIC experienced high rates of poverty, housing instability or homelessness, and unemployment. Given the prevalence of these social risks, community stakeholders involved in the study design advocated for the inclusion of measures related to social determinants of health, including traditional measures of social risk factor exposure and client perceptions of need and whether support was available. At the 6-month interval, outcomes related to being currently homeless or having multiple social risk factors for chronic homelessness improved more in the intervention arm than the control (described below) [16]. Noting improvements in mental-health-related wellness and in the social determinant of health-related outcomes, community stakeholders and research collaborators urged academic partners to account for these emerging themes and expand the focus of subsequent surveys with particular focus on life satisfaction, as related to quality of life (QOL), and clients’ perceptions of need and support related to the social determinants themes of housing and employment/financial/and legal (EFL) concerns.

As a result of this community engagement, the 1-year and 3-year follow-up surveys included items related to domains of QOL and survey elements assessing (1) participant report of being asked about social need and (2) participant perception of experiencing unmet needs related to housing and EFL concerns. At the 1-year interval, this survey was limited to participants having some services for depression at any agency, while the 36-month survey was expanded to the entire sample and is the focus of this paper.

This study and all procedures were approved as in accordance with the Declaration of Helsinki by the institutional review boards at RAND and participating agencies.

### 2.2. Communities, Agencies, Intervention, and Clients

The complete study (including agency recruitment, randomization, the development of interventions, and primary outcomes) has been described in detail elsewhere [16]. In brief, the Hollywood Metro and South Los Angeles communities represent two under-resourced areas within the larger Los Angeles County. These communities were selected based on need, established partnerships with the university study site, and in concordance with county-level service planning zones. Within these locations, a final sample of 95 agencies (including both healthcare (“formal”) and non-healthcare (“informal”) agencies) were enrolled. These agencies were block-randomized to two active interventions—Resources for Services (RS) or Community Engaged Planning (CEP). The RS intervention offered independent agencies a “toolkit”, developed with community input, of best practices in screening, referral, and treatment for depression as well as training from study staff on intended use. The CEP approach brought agencies together to review progress learn and iteratively enhance their use of the toolkit through collaborative learning and peer-agency engagement over 18 months (end of 2009–July 2011). Following agency randomization and training, clients were enrolled by survey staff not affiliated with their care, using eligibility criteria of English or Spanish speaking, age ≥ 18, Patient Health Questionnaire (PHQ-8) ≥ 10 (moderate to severe depression). Across participating agencies, survey staff recruited and enrolled 1322 eligible adults, 981 of whom participated in baseline and were eligible for 6- and 12-month follow-up surveys with data collection 2010–2012. Throughout the study, clients were free to choose sites for any services, and prior evaluation documented declining retention for services from initial programs of enrollment [18].

After 12-month follow-up and the end of the active intervention periods, there was an extension study follow-up at 36 months following baseline completion. Of 1004 participants from 89 programs eligible for 3-year surveys (enrolled; completed at least one prior survey; not refusing further follow-up or reported as deceased), 600 participated. From the sample of 1004 participants, 24 were deceased (2%), 10 refused to participate (1%), 3 were ill/incapable, and 367 (37%) not reached. With stakeholder input, the 36-month follow-up included client-centered perspectives, including perceptions of social needs and life satisfaction, the focus of this study.

### 2.3. Measures

#### 2.3.1. Dependent Variables

The outcomes for this secondary analysis follow two question stems: (1) being asked about social need and (2) perceptions of unmet social need over the 6 months preceding the survey, asking separately for: housing and for employment, financial, or legal (EFL) issues (employment, financial, or legal issues were asked as a group). The “asked about” item was a binary indicator of client perception of receipt of social need screening in either formal or informal sectors from any provider type (clinician, case manager, etc.). Client perception of “unmet social need” was determined through a 2-part question: (1) “did you feel you needed help” and, for clients who answered “Yes”, (2) “Did you get the help you feel you needed?” for each social need. Unmet social need was defined as perceiving a need that was not met versus either not having a need or having a need that was reported as met.

#### 2.3.2. Independent Variables

A range of independent variables were included from each wave of data (baseline, 6, 12, and 36 months), with variables overlapping each wave and some unique to waves due to community input as noted below. This analysis included measures related to the original CPIC study design, participant demographic factors, health-related factors (including clinical measures such as depression and health knowledge or stigma), quality of life-related outcomes, and patterns of service use. Additionally, for the analyses of the dependent variable “being asked about social need”, unmet need was included as an independent variable. While most co-variates in this analysis have been described elsewhere [16,18,19], several merit specific attention. For co-variates not described below, criteria for measurement and classification are included in Table 1 footnotes or in Appendix A, Table A1.

Life difficulties, aa measures of stressful events (15 items at baseline and 16 items at 6, 12, and 36 months), were assessed, including participant report of financial, housing, legal concerns/arrest, exposure to violence/trauma, loss in relationships (death or loss of child custody), and employment difficulties (e.g., loss of hours) in the 6 months prior to survey [20]. Barriers to care in the past 6 months (from a total of 17 barriers at 6 and 12 months; 19 barriers at 36 months) included participant report of challenges in obtaining needed care. This measure encompassed structural concerns (e.g., health insurance coverage or time to next appointment), stigma (fear or embarrassment), discrimination (race/ethnicity or legal status), and logistical factors (needing to take time off work or help to care for children) [21,22]. Life satisfaction, a strength-based evaluation, was included at 36-months due to stakeholder input in the community-participatory process. The original item was made up of an 8-item measure (overall life satisfaction as well as satisfaction related to physical health, mental health, work, housing, finances, area of residence, relationships) with responses from 0–10 (fully satisfied). For this analysis, we created a composite measure of satisfaction related to health-related social factors: social determinant of health satisfaction, representing the mean of the work, housing, finances, and area of residence subscales.

### 2.4. Analyses

We described sample characteristics assessed at baseline using means and standard deviations for continuous variables and percentages for categorical variables. We conducted univariate and multivariate logistic regression analyses to identify social demographic, clinical variables, and service utilization variables associated with being asked about social needs or unmet social needs. For being asked about social needs, we utilized a generalized estimating equation (GEE) framework [23] taking the correlation between two items (being asked about needed help with housing, being asked about needed help with finding employment or financial or legal issues) from the same person into account. Similarly, we fit GEE models for unmet social needs that defined a two-dimension outcome (unmet need to get help for finding housing, unmet need to get help for finding employment/financial/legal issues). Specifically, we used the SAS GENMOD procedure with a logistic link function assuming exchangeable correlation at the person level. Based on the estimated coefficients from the logistic regression model, for each predictor, we then conducted an overall F-test for the overall effect of that predictor on the two items simultaneously.

We developed a series of models including intervention status, social determinants, social context (problems, barriers), clinical and attitudinal factors, and service utilizations that may affect 36-month outcomes. First, we tested one variable at a time in univariate models for predictors at baseline, 6 months, 12 months, and 3 years. We identified significant predictors and, for each survey period, conducted multivariate models to identify unique predictors controlling for other factors. We then tested multivariate models adding baseline, 6-, 12-, and 36-month significant predictors. For variables with more than one specification (e.g., housing or homelessness risk, services use), we conducted alternative models to find best fits also informed by hypotheses. Using two “best fit” multivariate logistic regression models (limited and expanded), we examined predictors of report for being asked about social needs. Using this strategy, we developed a final set of models for each outcome using predictors emerging as significant at *p* < 0.05, from the sequential process plus controlling for intervention status as a key design variable. In the final model, significant predictors were all from the 36-month follow-up. Findings were iteratively reviewed with stakeholders for consistency of hypotheses with client experience, with recommendations to include being asked about needs as a predictor of perceived need. Multiple imputation was used to deal with missing data [24,25,26] for the analytic sample eligible for 36-month follow-up, excluding the deceased. Analyses were conducted in SAS version 9.4.

## 3. Results

### 3.1. Participant Characteristics at Baseline

For the analytic sample for 36-month follow-up (imputed to those eligible excluding the deceased), baseline demographic characteristics as well as social needs measures at 36 month follow are shown in Table 1. The study sample (*N* = 980) was 58% female, predominantly Black (46%) and Latino (41%), with a mean (SD) age of 45.4 ± 12.8 years. Income was below the poverty level for 74% of the sample. A total of 21% of the total sample was currently doing any work for pay, 53% reported chronic homelessness risk factors, and 54% were uninsured. The prevalence of 12-month depressive disorder by Mini-International Interview (MINI) [27] at baseline among follow-up participants was 62%, and 54% of the sample reported three or more chronic medical conditions. The mean (SD) PHQ-8 score was 15.0 ± 4.1, corresponding to moderate depressive symptoms.

### 3.2. Distribution of Social Needs Inquiry and Unmet Social Need

As shown in Table 1 (36-month data, with imputation to main analysis sample), 21% of participants endorsed being asked about housing needs, and 18% endorsed being asked about employment needs. Client report of being asked about social need (either housing or EFL) by any healthcare or non-healthcare agency did not vary by initial study arm. Of the total sample, 28% and 33% of participants reported an unmet need for housing and EFL, respectively (compared to either having no perceived need or having a perceived need that was reported as met within the 6 months prior to the survey).

### 3.3. Predictors of Being Asked about Social Needs

In Table 2, we present regression results by each outcome independently (housing or EFL) as well as combined in overall F-test (both housing and EFL simultaneously). In the expanded model, both total number of life difficulties and receiving mental health visits in the informal sector in the 6 months prior to the 36 month assessment were associated with being asked about housing (OR = 1.17, 95% CI = 1.05–1.32, *p* = 0.007; and OR = 3.48, 95% CI = 1.75–6.93, *p* = 0.002, respectively), EFL concerns (OR = 1.22, 95% CI = 1.07–1.39, *p* = 0.006 and OR = 2.65, 95% CI = 1.40–5.02, *p* = 0.006), and also significant in the overall test (both *p*-values < 0.05). Client report of satisfaction with social risk factor-related domains at 36 months was inversely associated with being asked about housing in the limited model (OR = 0.87, 95% CI = 0.80–0.94, *p* = 0.001) as well as the limited overall F-test (*p* = 0.004). In the expanded model, formal sector visits for depression care in the 6 months prior to the 36 months assessment were associated with being asked about EFL concerns (OR = 1.66, 95% CI = 1.01–2.75, *p* = 0.047). Neither unmet social need for housing nor unmet need for EFL at 36 months were associated with being asked about social need in the final expanded model. Intervention status (CEP vs. RS) was not significant (*p* > 0.05).

### 3.4. Predictors of Unmet Social Needs

Using the same iterative multivariate logistic approach, we assessed predictors of patient perception of unmet social need for both housing and EFL concerns (Table 3). As with asked-about models, final significant predictors were all from the 36-month survey. As mentioned previously, unmet social need is defined as having a need that is not met compared to all others (no need or having need that is met).

Neither clinical severity of depression (PHQ-8 score) nor unmet emotional distress needs at 36 months were associated with the perception of unmet social need for housing or EFL (*p*-values > 0.05). Life difficulty (stressful events at 36 months) was significantly associated with unmet housing and EFL concerns (OR = 1.18, 95% CI = 1.04–1.34, *p* = 0.013 and OR = 1.42, 95% CI = 1.22–1.65, *p* < 0.001, respectively) and significant in overall testing (*p* < 0.05). Total barriers to care in 6 months prior to 36 months were associated with EFL needs (OR = 1.17, 95% CI = 1.11–1.24, *p* < 0.001) and significant in overall F-testing (*p* = 0.01), but were not significantly associated with unmet housing needs. In both models, social determinant of health satisfaction at 36 months was inversely associated with perceived unmet housing and EFL needs (individually and in the overall test). Intervention status (CEP vs. RS) was not significant (*p* > 0.05).

## 4. Discussion

This exploratory study examined client-centered experiences of social need screening and subjective perception of unmet social need for minoritized and under-resourced adults with depression across multiple service sectors—both formal healthcare and informal social and community service agencies. Study participants had characteristics that made them likely to benefit from efforts to address social risk factors—adults with depression, the majority of whom reported income under the poverty threshold, chronic homelessness risk, and both a high incidence of medical co-morbidity and poor mental health quality of life [28]. As their recruitment took place across various service sectors—representing real-world service use for persons accessing agencies according to their varied perception of need and available resources—this study provides an opportunity to examine client-perceived social need across multiple sectors and institutions compared to studies that typically focus on single programs or institutions and focus on independently measured social factors rather than client views of being asked about or having needs.

Results from multivariate regression of social need inquiry shed light on the relevance of both life difficulty and patterns of service-sector use as predictors of being asked about social need. Additionally, we found that clients’ perception of unmet need was shaped by both individual-level hardships (life difficulties, barriers to care) and their life satisfaction (social determinant of health satisfaction). Underscoring the importance of multi-sector care systems for expanded collaborative care for depression, we found a positive association for clients accessing informal (non-healthcare) agencies for mental health concerns and being asked about social need. This pattern of results is consistent with our hypothesis that both individual and system factors would predict perceived unmet need and being asked about social needs. Similarly consistent with our second hypothesis, the final models for both perceived unmet need and being asked about social needs were all from 36 months (contemporaneous). However, some factors such as depression severity (PHQ-8) at 36 months were not significant predictors. Further, although being asked about social needs was tested as a predictor of unmet need given stakeholder input, it was not a significant predictor in final models. However, this may be because they were tested for the same time period and not in sequence, which is an issue for exploration in future studies.

In this study, approximately one-third of participants reported the perception of unmet social need for either housing or EFL concerns—a lesser proportion than those reporting homelessness risk or poverty on intake. This pattern, in which report of social need is less than prevalence of social risk factors, matches trends seen elsewhere. In their evaluation of social risk factors, Gold et al. reported findings of a community health center population where greater than 95% of participants identified social risk, while only approximately 20% of these clients requested help. They posit that this difference is partially explained by distinctions between social risk factors and client-identified social needs as well as clients’ readiness to accept social supports in healthcare settings [29]. This evaluation extends these observations—exploring which factors influence perceptions of social need and how willingness to accept support may vary by the context of inquiry: who asks about social need (healthcare providers or staff in informal sectors), the expectations of adults with depression in accessing formal vs. informal resources, and perhaps even the relationship of organizations with the community (operationalized in reported barriers to care) and resulting trust in recognizing or addressing social factors.

In contrast to social risk factor screening alone, this analysis suggests that care-as-usual that does not elicit client perceptions and priorities may be insufficient to address the underlying social need factors influencing health and quality-of-life outcomes. The findings draw attention to how the individual perspectives of clients with depression may be particularly salient in efforts to increase engagement regarding interventions for social risk [30,31,32]. This may have implications for services and policy as well as metrics for tracking outcomes in systems. As one example of an intervention utilizing individual client perspectives, O’Connell et al. demonstrated improved outcomes, including service use and housing tenure, when considering patient preferences and priorities in housing interventions [33,34]. This could reflect an area where emphasis on clients’ social needs may improve the delivery and utilization of interventions dedicated to addressing them—an important area for future research, particularly in under-resourced communities.

Limitations of this study include recruitment from only two local geographies and dependence on participant report and subsequent recall bias, which may factor into respondent recollection of both outcomes, as well as data from 2009–2014. Given the upsurge in initiatives to address social needs that have emerged since CPIC [35], the pattern of findings in this evaluation may no longer be representative of trends in screening patterns or patient report. However, this limitation is balanced by the study’s unique emphasis on client experience and perception of social need (as distinct from social risk), which addresses an important gap in current literature and extends contemporary social determinant frameworks. Point-in-time survey evaluation of social need limits our ability to understand changes in needs over time and our analysis cannot determine causation or the direction of association between predictors and outcomes, particularly given that predictors are from the same time period as outcomes. However, this may emphasize the importance of assessing a comprehensive view of social risk and need at the same time.

Although clinical measures of depression (PHQ) were not significantly associated with either outcome, prior studies have shown reciprocal relationships between mental health and social determinants [36]. Our study is limited to adults who experienced depression, so the pattern of findings may not be readily applied to populations without mental health needs and may be important to replicate for those with other conditions. Further data were generated prior to the COVID-19 pandemic, which has increased public and provider attention to social risk factors driving health inequities and increases in prevalence of mental distresses, anxiety, and depression [37,38,39].

While social risk interventions may successfully address population health and healthcare cost concerns [40], this analysis suggests that the incorporation of patient/client perception of social need—the subjective needs and priorities of individuals—may also be important to achieving client-centered care in relation to health-related social factors [41]. Tailoring services in accordance with client choice may be a means to increase not only the efficiency of mental health service delivery but also its effectiveness. Some frameworks of health behavior emphasize individual-level determinants, such as perceived health and personal health practices, in determining health services behavior [42]. This study demonstrates the context-dependent nature of client perception, for example, the site of service for social need screening (healthcare vs. non-healthcare). These finding suggest that client decision making may be influenced by factors at multiple upstream social-ecological levels (e.g., a client with a child may underreport housing needs in healthcare settings due to concerns about triggering a child protective services investigation)—not all of which could be evaluated by this study but which point to avenues for future investigation and responsive organizational policies [17,43].

Organizational responsiveness may be prompted by external factors including but not limited to value-based transitions in financing [44] and may also require systems redesign—including trainings to increase provider preparedness, systematic data collection on clients’ social needs, a more robust IT infrastructure to track outcomes while simultaneously ensuring equity, and increased systems capacity to sustain viable solutions for social needs within the healthcare organization or through strategic community partnerships [6,45,46,47].

## 5. Conclusions

While developing partnerships and soliciting client feedback may optimize delivery of social needs interventions, these efforts do not necessarily address broader concerns of process that generate racialized resource scarcity in communities of color and, in turn, poor health outcomes [48]. Nonetheless, healthcare providers and organizations, together with community-based partners, are frontline witnesses to health effects of social needs in their patients and communities. As such, they share responsibility to advocate for change in the processes that generate the income inequity, housing segregation, unaffordability, and scarcity as well as toxic food environments that are up-stream drivers for social needs in minoritized populations and to track the individual and community risks as well as client perceptions of need to support engagement for improved outcomes. Alongside transformative social and policy changes to address poverty and structural racism, partnered approaches involving healthcare and community services to generate client-centered survival programs with client and community engagement and tracking of perceptions of need and support are urgently needed with evaluation to inform policy.

## Figures and Tables

**Table 1 ijerph-19-10212-t001:** Baseline Characteristics (and for social needs, 36-month survey) for Analytic Sample (Imputed) for 36 month follow-up for Community Partners in Care (*N* = 980) ^a^.

Characteristics	*N*	%	M ± SD
**Baseline Assessment and Screening ^b^**			
Intervention arm			
RS	483	48.6	
CEP	497	51.4	
Female sex	581	57.9	
Age, years			45.4 ± 12.8
Race/Ethnicity			
Latino	396	41.2	
African American	469	46	
Non-Hispanic white	81	9	
Other	34	3.7	
Less than high school education	430	43.8	
Income under poverty level	723	73.8	
No health insurance	525	54.1	
≥3 chronic health conditions from list of 18	521	54	
12 month depressive disorder (MINI, baseline)	605	61.8	
**Baseline, Repeated Survey Measures ^c^**			
Married or living with partner	223	22.7	
Doing any work for pay at the present time	203	20.5	
Chronic homelessness risk ^d^	514	53.6	
Alcohol abuse or illicit drug use, prior 12 months	383	39.4	
Poor mental health quality of life	530	53.8	
Number of life difficulties from a list of 15			4.1 ± 2.8
PHQ-8 ^e^			15.0 ± 4.1
MCS-12 ^f^			39.2 ± 7.3
PCS-12 ^g^			39.4 ± 7.2
**36-Month Survey of Social Needs ^h^**			
Asked about housing needs	201	20.5	
Perceived unmet need for housing	278	28.4	
Asked about E/F/L ^i^ needs	178	18.4	
Perceived unmet E/F/L needs	323	33.0	

^a^ Data were multiply imputed and weighted for eligible sample for 36-month follow-up excluding deceased; N, unweighted, %, or mean weighted. ^b^ Measured only at the time of enrollment to study (baseline). ^c^ Measured repeatedly in Community Partners in Care (CPIC) (see Appendix A, Table A1). ^d^ Homeless or living in a shelter or at least two of four risk factors (at least two nights homeless, food insecurity, eviction, or financial crisis). ^e^ PHQ, Patient Health Questionnaire; possible scores range from 0 to 24, with higher scores indicating greater depression severity. ^f^ MCS, Mental Component Summary (of the Short-Form 36); possible scores range from 0 to 100, with higher scores indicating better physical health. ^g^ PCS, Physical Component Summary (of the Short-Form 36); possible scores range from 0 to 100, with higher scores indicating better mental health-related quality of life. ^h^ Primary outcomes measured at 36 month and in relation to social need in the preceding 6 months. ^i^ E/F/L, employment/financial/legal issues. Abbreviations: RS, Resources for Services; CEP, Community-Engaged Planning; MINI, Mini Neuropsychiatric International Interview.

**Table 2 ijerph-19-10212-t002:** Models for being asked about housing and employment financial and legal issues at 36 months, by perceived life difficulties and service utilization controlling for intervention condition.

Variables	Model 1							Model 2						
	Being Asked about Housing			Being Asked about Employment Financial and Legal Issues			Omnibus Test	Being Asked about Housing			Being Asked about Employment Financial and Legal Issues			Omnibus Test
	OR	95% CI	*p*	OR	95% CI	*p*	*p*	OR	95% CI	*p*	OR	95% CI	*p*	*p*
Intervention arm, CEP vs. RS	0.89	(0.50–1.58)	0.677	0.70	(0.37–1.34)	0.251	0.494	0.95	(0.54–1.69)	0.864	0.70	(0.36–1.34)	0.246	0.473
PHQ8 ≥ 10	1.12	(0.50–2.48)	0.766	0.54	(0.24–1.23)	0.127	0.197	1.10	(0.48–2.50)	0.803	0.54	(0.24–1.22)	0.12	0.199
Total number of life difficulties	1.20	(1.09–1.33)	<0.001	1.24	(1.10–1.41)	0.002	<0.001	1.17	(1.05–1.32)	0.007	1.22	(1.07–1.39)	0.006	0.004
SDoH Satisfaction	0.87	(0.80–0.94)	0.001	1.00	(0.93–1.08)	0.98	0.004	0.92	(0.82–1.02)	0.118	1.01	(0.94–1.10)	0.723	0.187
Any visit in formal sectors for Depression care	1.55	(0.69–3.51)	0.252	1.56	(0.97–2.52)	0.068	0.222	1.47	(0.71–3.04)	0.266	1.66	(1.01–2.75)	0.047	0.169
Any visit in informal sectors for MH	3.18	(1.76–5.74)	<0.001	2.72	(1.41–5.24)	0.006	<0.001	3.48	(1.75–6.93)	0.002	2.65	(1.40–5.02)	0.006	0.001
Unmet for finding housing								2.86	(1.24–6.58)	0.019	1.04	(0.60–1.80)	0.876	0.015
Unmet for finding employment/financial or/legal issues								0.75	(0.43–1.32)	0.309	1.44	(0.93–2.22)	0.099	0.147

Multivariate logistic regression from GEE model with predictors listed in the first column. Odds ratios (OR) with 95% confidence intervals (CI). Abbreviations: CEP, Community-Engaged Planning; RS, Resources for Services; PHQ, Patient Health Questionnaire; SDoH, social determinants of health; MH, mental health.

**Table 3 ijerph-19-10212-t003:** Models for perceived unmet needs for housing assistance and employment financial and legal assistance at 36 months, by perceived life difficulties, life satisfaction, barriers to care, and service utilization controlling for intervention condition.

Variables	Model 1							Model 2						
	Unmet for Housing			Unmet for Employment Financial and Legal Issues			Omnibus Test	Unmet for Housing			Unmet for Employment Financial and Legal Issues			Omnibus Test
	OR	95% CI	*p*	OR	95% CI	*p*	*p*	OR	95% CI	*p*	OR	95% CI	*p*	*p*
Intervention arm, CEP vs. RS	0.82	(0.48–1.41)	0.437	1.22	(0.84–1.80)	0.292	0.379	0.81	(0.48–1.36)	0.404	1.22	(0.83–1.81)	0.304	0.347
PHQ8 ≥ 10	0.79	(0.49–1.27)	0.323	0.77	(0.39–1.51)	0.41	0.491	0.82	(0.50–1.35)	0.418	0.76	(0.35–1.67)	0.446	0.567
Unmet for receiving help you feel you need for emotions	1.32	(0.88–1.99)	0.169	1.69	(0.88–3.26)	0.103	0.11	1.36	(0.88–2.11)	0.166	1.44	(0.75–2.76)	0.245	0.26
Any MH outpatient visits	1.34	(0.77–2.32)	0.275	0.53	(0.33–0.85)	0.011	0.03	1.31	(0.74–2.32)	0.326	0.54	(0.37–0.81)	0.003	0.025
Went to social service agency to get assistance	1.25	(0.85–1.83)	0.261	1.45	(0.99–2.11)	0.055	0.103	1.45	(0.96–2.19)	0.079	1.84	(1.24–2.73)	0.003	0.006
SDoH Satisfaction	0.71	(0.62–0.83)	<0.001	0.84	(0.78–0.90)	<0.001	<0.001	0.69	(0.59–0.80)	<0.001	0.79	(0.72–0.87)	<0.001	<0.001
Total number of life difficulties	1.18	(1.04–1.34)	0.013	1.42	(1.22–1.65)	<0.001	<0.001							
Total number of barriers of getting MH care								1.04	(0.94–1.14)	0.394	1.17	(1.11–1.24)	<0.001	0.001

Multivariate logistic regression from GEE model with predictors listed in the first column. Odds ratios (OR) with 95% confidence intervals (CI). Abbreviations: CEP, Community-Engaged Planning; RS, Resources for Services; PHQ, Patient Health Questionnaire; MH, mental health; SDoH, social determinants of health.

## Data Availability

Not applicable.

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
