# Peer review of "Multi-Sector Assessment and Client-Perception of Social Need at Long-Term Follow-Up of a Group-Randomized Trial of Community-Engaged Collaborative Care for Adults with Depression"

_ijerph, 2022, doi:10.3390/ijerph191610212_

Round 1
Reviewer 1 Report
I have completed my evaluation of a manuscript titled “Multi-Sector Assessment and Client-Perception of Social Need at Long-term Follow-up of a Group-Randomized Trial of Community-Engaged Collaborative Care for Adults with Depression”.
This version of the article can accept in its present form.
This manuscript is a resubmission of an earlier submission. The following is a list of the peer review reports and author responses from that submission.
Round 1
Reviewer 1 Report
I have completed my evaluation of a manuscript titled “Multi-Sector Assessment and Client-Perception of Social Need at Long-term Follow-up of a Group-Randomized Trial of Community-Engaged Collaborative Care for Adults with Depression”.
The study is well written and based on a relevant topic for mental health in the community, and specialy for people with depression. The comments for improving the article are below.
Introduction
The introduction is informative and well written. I do however miss the clear aim. Although the authors defined research questions, it would be improved if the authors write the clear aim.
Who are “institutional review boards” ? Are they Ethical committee? It should be write that this study was performed in accordance with the Declaration of Helsinki ethical standards. So, plaese notice this. Please be precise and give the number of approvals.
Response: Institutional review boards are the human subjects committees of organizations. We include that they follow Helsinki standards. We clarified that the review was conducted and approved by RAND as the lead organization, and also at 7 participating agencies that required review by their own IRB or research review committee.
Materials and Methods
Although the authors describe the protocol it would be improved if the authors specify the number of enrolled participants.
Response: The number of enrolled participants is provided in the description, both initially and the number eligible for follow-up periods. We have included the term enrolled in addition to recruited to make this clear.
Tables
The aberrations RS, CEP, PHQ, MSC, PSC, SDoH, MH, CPIC, etc need to be named in footer of table
Response: This is clarified in each table, and the abbreviations are also defined in the text.
In line132 is tipfeler (..)
Response: This is removed.
Reviewer 2 Report
This is a very interesting paper that needs to address some important issues before publication.
Abstract: Objective: to examine perceptions of “social needs” related to housing and employment, financial, and legal (EFL) concerns among individuals with depression (Lines 33 and 34). Please specify when, Is it at baseline, 6, 12, or 36 months? Is it with or without intervention?
Response: This is perception of social need for all participants (those with RS as well as CEP) at 36 months; this is clarified in the abstract.
Introduction.
- There is no objective. Authors state research questions (Lines 91 to 95). Please specify when, is it at baseline, 6, 12, or 36 months? Is it with or without intervention?
Response: The introduction has been revised to reflect the objective of the study and clarify time point for data used in analysis.
2. Lines from 97 to 102 should be changed to the material and method section
Response: Materials and methods. (noted at line 131 now)
Material and methods.
1. How are the Community Partners in Care recipients selected?
Response: CPIC agencies were selected according to criteria described in lines 140-145. These agencies were randomized to RS vs CEP interventions. Individual participants were recruited by RAND survey staff not involved in the care of patients, according to criteria described in lines 173-186.
2. How was depression diagnosed? who made the diagnosis? With what criteria? what treatment is received by depressive individuals? what follow-up was offered to individuals with depression?
Response: As described in lines 173-186, individual participants were screened for depression by survey staff using the PHQ-8, and with inclusion criteria of PHQ-8>=10 (for moderate to severe depression).
In relation to treatment, that was up to the care practices of the agency and client preferences. However, agencies were supported in best practices to both study intervention conditions. For the control condition, individual agencies received individual trainings in resources for collaborative care enhanced for community organizations; while in the active intervention condition, a multi-agency collaborative intervention approach with adaptations suggested by partners was used to support cross-agency training and collaboration. Within both conditions, then participating patients received services based on the practices followed by the agencies. The study itself did not assign a particular treatment to individuals. This paper is not focused on intervention comparisons but rather the social needs assessment at long-term follow-up.
3. Confusion on selection. The authors mention randomization of “agencies” and suddenly state: “The study recruited 1,322 eligible adults, 981 of whom participated in baseline and were eligible for 6- and 12-month 154 follow-up surveys with data collection 2010-2012” (lines 153 and 154) with or without intervention? (Apparently, all had an intervention, please clarify)
Response: Thank you for opportunity to clarify – Randomization took place at level of agencies, and participating agencies were the target of intervention. In turn, survey staff from the study recruited participants within participating sites, who then accessed services at their site. This portion of the manuscript has been revised to reflect this process. The study had two active intervention conditions – individual agency support for collaborative care resources; or community engagement and planning for collaborative support across agencies in that condition, for collaborative care. The “with or without intervention” is referring to the “active” collaborative condition but we have clarified this.
4. What did the intervention consist of?
Response: As described in lines 166-172, the RS intervention provided individual agencies with a toolkit of best practices in depression care with trainings and resources but for individual agencies. The CEP intervention provided the same toolkit enhanced by collaborative and community-engaged planning, training, and follow up implementation.
5. Authors asked: “did you feel you needed help” (line 174). Who is supposed to give this help?
Response: This was up to the perspective of the participant; it could have been through a formal provider such as healthcare or an informal source such as a faith-based or other agency for support. It was specifically defined broadly given the broad range of agencies included in the study.
6. What are the selection criteria to be a recipient of the program?
Response: As described in 159-165, agencies were selected on basis of proximity to areas of concentrated need, according to county-level services areas (the basis for health services delivery in Los Angeles County) and established ties to University partnership. Greater detail regarding agency recruitment is included in the primary study, as referenced.
7. What are the selection criteria to be included in the exploratory secondary analysis?
Response: As described in lines 189-193, eligibility for participation in the 36 month telephone survey included continued enrollment, completion of at least 1 prior survey, no prior refusal for continued participation, and not previously reported as deceased. Also as described, of the 1004 eligible participants, 404 did not participate.
8. The outcome/question about social need: Participants enrolled in CPIC already experience high rates of poverty, housing instability or homelessness, and unemployment (to be in the program). So why ask them again? Or the analysis corresponds to the initial assessment for being accepted in the Program? What happens if they objectively experience poverty, housing instability or homelessness, and unemployment but they did not perceive the need? Apparently, the analysis focuses on the 36-month follow-up. Please clarify.
Response: Correct, the analysis explores associations between being asked about social need and perceptions of social need at 36 months relative to certain demographic factors (measured at baseline) as well as other factors (measured repeatedly or in follow-up surveys). The participants are not specifically recruited for housing instability or other social needs, though agencies were included providing services for these issues as well as more general agencies such as healthcare settings. The study had an effect on some social determinants such as exercise and risk factors for homelessness at 6-month follow-up, and over the long-term, the community-partnered research team was interested, particularly community partners, in the experiences of individual participants around their social needs. This was seen as an important enhancement to asking for example, about housing status without asking for participant perception of their needs. Specifically, exploring the intersection between “objectively experienced” and “individual perception” is the basis for the study and is consistent with contemporary frameworks related to social drivers of health outcomes. See, for example, references by Alderwick (7) and Eder (10) In particular, “social needs, meanwhile, are not necessarily synonymous with social risk factors—they also depend on people’s individual preferences and priorities. Distinguishing between social risks and social needs emphasizes the patient’s role in identifying and prioritizing social interventions. This concept is at the heart of efforts to implement shared-decision making in traditional medical care” (Alderwick 412)
9. How was the depression at baseline, 6, 12, and 36 months?
Response: Depression was measured using the PHQ at baseline and follow-up.
10. They state, “analysis included measures related to the original CPIC study design … demographic factors, health-related factors, quality of life… and patterns of service use” (lines 182 to 184). And what about depression status?
Response: Yes, depression status is included in “health-related factors (including both clinical measures as well as health knowledge or stigma)” and we have now included “such as depression” with health-related factors (line 214).
11. Not clear how is the link between the independent variables such as quality of life-related, patterns of service use, and life satisfaction (among others) and the dependent variable unmet social need (housing/employment/ EFL concerns). It could be the other way around. If the social need was not met, there could be consequences on quality of life and health services use.
Response: Yes, correct. The purpose of this analysis is to explore associations between measured variables and outcomes of interest, especially as 3 year follow-up had a particular focus on these social need measures. Our study is not designed to make claims of causation. The analysis had the advantage of longitudinal data – but the strongest associations are for the same time period. The limitations related to causation are noted in the discussion (lines 422-423).
RESULTS
- “The prevalence of a depressive disorder at 12 months among participants was 62%” (lines 247-248). And at baseline? 6 and 36 months?
Response: While the PHQ8 was used as criteria for selection for the study, in addition at baseline the MINI diagnostic measure was administered; the variable is for the 12 months prior to baseline and is a baseline measure. We have clarified this in the text and Table 1.
2. Confusion. The authors state “Baseline demographic characteristics and survey measures at the time of randomization are shown in Table 1” (lines 242-243). But Table 1 says “Characteristics at 36 months follow-up of participants ….” (Wrong title, please correct)
Response: We have corrected the title and clarified in the text. The Table provides the descriptive baseline data as well as for social needs the 36 month data, for the main analytic sample, which has imputed data to apply to those eligible for 36 month follow-up minus deceased.
3. Table 1. “Repeated Survey Measures” subtitle. Please specify the time. Is it 6, 12, or 36 months?
Response: This is baseline data; we differentiate between measures from the screener or only in baseline versus those asked in all follow-ups, but all of the data except social needs measures in this Table 1 are from baseline. We have clarified this.
4. Table 1. Results focus on the 36 Month Survey. If this is the case, please specify this at the beginning and in the material and methods section. Nonetheless, it also needs to specify if the information is with or without intervention (not clear).
Response: This paper concerns analysis of data across intervention conditions. Intervention is a design variable that is not significant, and this analysis focuses on the broader framework of predictors from different time periods and “best fit” analysis results using data across different waves. Each Table includes intervention status (as a design variable) and we added a sentence inmain results section that intervention status is not significant. The focus on 36-month data is in the introduction line 97; and we clarified this focus and justification (as it’s the only wave where these issues were asked of the whole sample) lines 153-154.
5. Table 2. Why is the unmet social need a predictor of being asked about housing? I believed unmet social need was proposed as a dependent variable.
Response: It was included as an additional predictor because it is important to know if providers respond in asking about social needs like housing, when the client perceives that there is an unmet need – does that cue the provider to ask. Since the significant predictors are largely from that time frame, it was viewed by stakeholders as an additional variable to include. This is clarified in lines 281-282. However, it was not significant in the final model and this is now clarified in the discussion, lines 408-410.
CONCLUSIONS.
1. Authors do not respond to the hypothesis. They say: “We hypothesized that both patient- (including, but not limited to, individual clinical symptoms, service use patterns, experiences of barriers to care, report of lifetime stressful events, and measures of recent life satisfaction) and system-level factors would predict report of being asked about social needs and perceptions of unmet social need at follow-up (3 years)”. But they did not provide the answer ….
2. They also say, “We hypothesized that experiences in the 6 months prior to the assessment at 3 years would be particularly predictive of reported social needs”. But they did not provide the answer ….
Response: For both points (1 and 2 above) related to the conclusions, we have modified the discussion to refer to how results relate to these hypotheses, lines 403-408.
Reviewer 3 Report
Please see the attached file.
